# The FLA4-FEI Pathway: A Unique and Mysterious Signaling Module Related to Cell Wall Structure and Stress Signaling

**DOI:** 10.3390/genes12020145

**Published:** 2021-01-22

**Authors:** Georg J. Seifert

**Affiliations:** Institute of Plant Biotechnology and Cell biology, University of Natural Resources and Life Science, Muthgasse 18, A-1190 Vienna, Austria; georg.seifert@boku.ac.at

**Keywords:** SOS5, receptor-like kinase, cell wall integrity, ACC, arabinogalactan protein, root development, seed coat mucilage, pectin, cellulose

## Abstract

Cell wall integrity control in plants involves multiple signaling modules that are mostly defined by genetic interactions. The putative co-receptors FEI1 and FEI2 and the extracellular glycoprotein FLA4 present the core components of a signaling pathway that acts in response to environmental conditions and insults to cell wall structure to modulate the balance of various growth regulators and, ultimately, to regulate the performance of the primary cell wall. Although the previously established genetic interactions are presently not matched by intermolecular binding studies, numerous receptor-like molecules that were identified in genome-wide interaction studies potentially contribute to the signaling machinery around the FLA4-FEI core. Apart from its function throughout the model plant *Arabidopsis thaliana* for the homeostasis of growth and stress responses, the FLA4-FEI pathway might support important agronomic traits in crop plants.

## 1. Introduction

A relatively new field of signaling research in plants was arguably kicked off by the publication of an article on the crucial importance for cell wall integrity (CWI) for many aspects of plant life [1]. CWI signaling has been genetically dissected in yeast [2] and the central importance of cell-to-extracellular matrix (ECM) interactions in human diseases has driven related research in mammalian cells [3,4,5]. In plants, the enormous structural and developmental heterogeneity of cell walls pose particular challenges to the investigation of CWI signaling, however there already exist numerous receptor molecules and signal transduction elements that were proposed to act in this process [6,7,8,9,10,11]. In most cases, however, a clear definition of what is actually sensed and how this information is transduced is still lacking, which is not surprising given the difficulty to experimentally manipulate cell walls without causing massive pleiotropic side effects. Also, the term CWI sensing is sometimes overenthusiastically applied to an observed phenomenon involving mutations in a receptor-like molecule that somehow affect cell walls. However, similar to CWI signaling in yeast and ECM signaling in animals, plant CWI-signaling modules most likely act both outside-in (to sense the mechano-chemical status of the cell wall) as well as inside-out (to modify the cell wall function according to developmental needs and stress). Based on genetics alone we can only speculate on the true role of a protein in a molecular process. However, at present, to *directly* assay cell wall sensing in planta is still elusive. With this note of sobriety, a shortlist of *Arabidopsis thaliana* loci encoding receptor-like molecules previously implicated with CWI includes *WALL ASSOCIATED KINASE 1 (WAK1)* and its paralogs [12]; *THESEUS1 (THE1), FERONIA (FER)* and other Malectin-like receptor kinases [13]; *BRASSINOSTEROID INSENSITIVE 1* (*BRI1*) and *RECEPTOR-LIKE PROTEIN 44* (*RLP44*) [14,15]; *MALE DISCOVERER INTERACTING LOCUS 2* (*MIK2*) [16]; *STRUBBELIG* (*SUB*) [17]; as well as *FEI1* and *FEI2* (see below). Together with *FASCICLIN-LIKE ARABINOGALACTAN PROTEIN 4 (FLA4)* also known as *SALT OVERLY SENSITIVE 5 (SOS5)*, the two *FEI* loci form the core components of a putative signaling module apparently involved in CWI maintenance throughout the plant. For the first time in a dedicated review, I will focus on the *FLA4-FEI* pathway in an attempt to summarize presently available evidence into an updated working model of its role for CWI control.

## 2. *FLA4* and the Partially Redundant *FEI1* and *FEI2* Loci Act in a Linear Genetic Pathway Affecting Both Cellulose and Pectin

The identification of the *FLA4-FEI* pathway as a potential CWI signaling module was the consequence of a series of serendipitous discoveries. A forward genetic screen targeted at the elucidation of salt tolerance isolated several *salt overly sensitive (sos)* mutants in *A. thaliana*. While the *SOS1*, *SOS2,* and *SOS3* loci that encode a Na^+^/H^+^ antiporter, a protein kinase, and a Ca^2+^ binding protein, respectively, were elegantly related to the immediate cellular response to excessive salt [18], the *SOS5* locus was found to be allelic to *FLA4* encoding a member of the family of fasciclin-like arabinogalactan proteins (FLAs) [19]. The mechanistic relation of *FLA4* to salt tolerance is still unclear. However, the predicted localization at the plasma membrane-to-apoplast interface and the dramatic radial expansion of roots of *fla4* mutant seedlings growing on media containing 100 mM NaCl [20] was reminiscent of conditional mutants with defective primary cell walls [21,22]. Moreover, *fla4* roots displayed abnormal cell wall thickness and cell adhesion [20]. Thus, on the formally genetic level, *FLA4* is required for the normal architecture and function of cell walls.

All FLAs contain one or two fasciclin 1 (Fas1) domains. Fas1 domain-containing proteins occur in all phyla of life including bacteria, fungi, plants, and animals and they are involved not only in (mechanical) cell adhesion but in various roles related to the interaction of cells with their environment. For instance, the human Fas1-protein periostin acts in both outside-in and inside-out signaling by mechanically connecting various cell types to ECM components via integrin complexes [23] and as cytokine-like paracrine signals establishing the metastatic niche of human glioblastoma [24]. However, despite considerable progress in determining structures of different Fas1-containing proteins and numerous discoveries related to their action in humans, fungi, bacteria, and plants, presently there is no comprehensive mechanistic model of the molecular action of any representative of this superfamily. Notwithstanding the lack of a mechanism for Fas1-proteins, it can be said that their ancestral function best fits the definition of matricellular proteins that act in the extracellular matrix, often combining biomechanical roles with biochemical signaling functions (reviewed in [25]). All available evidence points to a corresponding complex role of FLA4 in the FLA4-FEI pathway.

The two other core components of the FLA4-FEI pathway were discovered during a long-time investigation of the control of ethylene biosynthesis, with a focus on 1-AMINOCYCLOPROPANE 1-CARBOXYLIC ACID (ACC) SYNTHASE (ACS) isoforms [26]. Two homologous leucine-rich receptor-like kinases (LRR-RLKs) were found to interact with two ACS isoforms in a yeast two-hybrid assay. While the phenotypes of single mutants of each of the two *FEI* loci initially were reported to be unremarkable, double mutants showed a subtle ectopic radial expansion in different plant organs, which in roots of seedlings growing on 4% sucrose-containing media was dramatically enhanced. Fittingly, the loci were named *FEI1* and *FEI2* for ‘fat’ in Chinese [27]. In addition to the conditional root swelling phenotype, the *fei1 fei2* double mutants were oversensitive to the cellulose synthase inhibitor isoxaben and showed a reduction in crystalline cellulose. Due to the fact that these symptoms were previously described for mutant alleles in several other loci, combinations between *fei1 fei2* and *cobra (cob)* [28], *cesa1/radial swelling 1* [29], *cesa6/procuste1* [30], and *fla4* [20] were generated. Although all the tested mutants showed sugar-sensitive root swelling similar to *fei1 fei2,* only the triple mutant *fei1 fei2 fla4* showed non-additivity, while the other triple mutants displayed a synergistic enhancement of the *fei1 fei2* phenotype even under low-sugar conditions. This observation suggested that *FLA4* and the *FEI* loci acted in a linear genetic pathway that was synergistic with, but independent of, primary cell wall cellulose biosynthesis mediated by the *CesA1*, *CesA6,* and *COB* loci.

The action of the *FLA4-FEI* pathway was found not only to affect rapidly elongating roots and stem thickness [27], but also the development of seed coat mucilage. During seed maturation, the epidermal cells of the seed coat produce a specialized cell wall that encapsulates a pocket of highly hydrophilic polysaccharides that rapidly swell upon hydration and coat the seed with a sticky mucilaginous layer, adhering to the remnants of the burst cellulosic primary cell wall. While seed coat mucilage is not essential for plant survival, the abundance, adhesion, and composition of seed coat mucilage can be easily assayed. Consequently, in genetic screens for seed coat mucilage-defective mutants, loci involved in the biosynthesis and remodeling of pectin, cellulose, and hemicellulose have been found [31,32]. Therefore, the finding that *FLA4* and *FEI2* were required for normal seed coat mucilage [33] was an important milestone, not only to better understand the genetic interactions between the FLA4-FEI pathway and cell wall polysaccharides, but also to realize that the FLA4-FEI pathway acted in CWI maintenance throughout the entire plant.

## 3. The *FLA4-FEI* Pathway Genetically Interacts with Primary Cell Wall Biosynthesis and Damage at Various Levels

The primary cell wall is deposited while cells expand and it consists of three predominant polysaccharides: cellulose, hemicellulose, and pectin. Cellulose microfibrils are deposited perpendicular to the main axis of cell expansion and they comprise a major mechanical load-bearing structural element, while hemicellulose and pectin interact with cellulose and with each other to regulate the controlled deposition and relaxation of growing primary walls [34]. The precise modes of these interactions still are mostly elusive, but an emerging model proposes the coordination of the three major cell wall polysaccharides in biomechanical hotspots [35]. While cellulose and hemicellulose on the one hand and pectin on the other hand have previously been depicted as forming largely independent networks in the cell wall [36,37], the biomechanical hotspot model suggests a close association between all three groups of biopolymers in biomechanically critical microdomains [38]. A recent example of the physiological link between pectin and cellulose was the demonstration that the *QUASIMODO2* (*QUA2*) locus, which primarily acts in pectin biosynthesis, also had a great impact on cellulose biosynthesis and crystallinity [39]. Moreover, to reinforce the cell wall in situations of biotic and abiotic stress, crosslinking of cell wall glycoproteins, such as extensins and arabinogalactan proteins, by cell wall peroxidases might play an important conditional role [40,41,42]. As will be explained in this article, I propose that the FLA4-FEI pathway might co-ordinate between the cellulose/hemicellulose/pectin network of primary cell walls and oxidative cell wall crosslinking.

When the combined phenotypic evidence gathered from *fla4* and *fei2* single and *fei1 fei2* double mutants is considered, the observations are comparable with mutants in pectin biosynthesis, such as abnormally thin middle lamella and disrupted cell adhesion. However, the mutants also displayed reduced cellulose crystallinity, hypersensitivity to isoxaben, and radial expansion, which are typical for cellulose biosynthetic mutants. Moreover, it was shown that the transcript levels of *FEI1* and *FEI2* were significantly reduced in pectin-deficient *qua2* mutants [39], which was interpreted as evidence of an important role of pectin in controlling cellulose deposition. Furthermore, it was shown that loss of function mutations in the *SHOU4* and *SHOU4L* loci (shou meaning ‘slim′ in Chinese) that encode novel membrane proteins partially suppressed the *fei1 fei2* phenotype and increased cellulose deposition. The SHOU4 proteins were suggested to negatively control cellulose deposition, probably by regulating trafficking of cellulose synthase, and the additive interaction of *shou4* mutants with *fei1 fei2* supported the importance of both the FLA4-FEI pathway and the SHOU4 loci for cellulose deposition in separate antagonistic pathways [43].

Thorough genetic dissection of the role of *FLA4* and *FEI2* loci for seed coat mucilage adhesion suggested a relation of the *FLA4-FEI* pathway with pectin rather than cellulose biosynthesis [44,45]. The central observation in these studies was the non-additivity of the *fla4* mutant phenotype with seed coat mucilage-defective *mucilage modified 2* (*mum2)* and *flying saucer 1* (*fly1)* mutants that both directly affect pectin properties. *MUM2* encodes an α-galactosidase that acts on galactan side chains of pectin and its loss-of-function mutation blocked the extrusion of mucilage upon hydration. This block was reverted in the *fla4 mum2* double mutant. By contrast, *FLY1* encodes an E3 ligase regulating pectin methyl esterification—in *fly1* mutants the seed coat mucilage remained abnormally adherent to the primary cell wall, resulting in ‘UFO′-shaped appendages. Also this ectopic adhesion phenotype was reverted in the *fla4 fly1* double mutant, both results suggesting a role of *FLA4* for pectin structure that influences adhesion counteracting *MUM2* and *FLY1* [44]. The *CesA5* locus is involved in cellulose synthesis in the seed coat primary cell wall and the *cesa5* mutant showed a loss of mucilage adhesion similar to *fla4*. The *fla4 cesa5* double mutant phenotype was more severe than either single mutant suggesting that each locus affected cell wall structure independently from each other. The *fei2* mutant showed a mucilage phenotype similar to *fla4* and *cesa5,* but the *fei2 cesa5* double mutant was more severe than the single mutants, while the *fla4 fei2* double mutant appeared identical to the *fla4* and *fei2* single mutants. This placed *FLA4* and *FEI2* in a linear genetic relation to control pectin structure in seed coat mucilage, independent of cellulose biosynthesis. Consistently, the *FLA4-FEI* module acted in the establishment of normal seed coat mucilage independently of *COBL2,* a genetic locus believed to directly influence cellulose biosynthesis in the seed coat [46].

Interestingly, the FLA4-FEI pathway was also shown to interact non-additively with two galactosyltransferase-encoding loci, *GALT2* and *GALT5,* required for the biosynthesis of the type II arabinogalactan structures that decorate numerous FLAs and other AGPs [47]. The phenotype of the *fei1 fei2 sos5 galt2 galt5* quintuple mutant and the *fla4* single mutant were identical with respect to both root growth and seed coat mucilage adhesion. A possible interpretation is that galactosylation of FLA4 by GALT2 and GALT5 is essential for FLA4 to function [47]. However, given that FLA4 is only lightly *O*-glycosylated and the *galt2 galt5* mutation leads to a considerable general reduction in AGP level, another possibility should not be excluded. In principle, the FLA4-FEI pathway might not depend on *O*-galactosylation of FLA4, but *O*-galactosylation of AGPs might control an unknown component of the FLA4-FEI pathway. This interpretation is consistent with the demonstration that an engineered *FLA4* variant lacking the putative *O*-glycosylation sites remained genetically functional [48]. In summary, it can be said that genetic analysis of the role of FLA4-FEI pathway for CWI showed its close interaction with pectin and AGP function and indirect importance for cellulose structure.

Yet another biological process involving the FLA4-FEI pathway is the response to cell wall damage (CWD). Inhibition of cellulose biosynthesis in *A. thaliana* by isoxaben or exposure to cell wall polysaccharide-degrading enzymes triggered a set of stress responses, including increased jasmonic acid (JA) and salicylic acid (SA) levels and signaling and ectopic lignification. To identify genetic loci possibly involved in this well-quantifiable CWD response, a several mutant loci previously implicated with cell wall signaling were tested. The study included mutant alleles of *THE1*, of *MID1 COMPLEMENTING ACTIVITY* (*MCA1)* encoding a putative mechanosensitive Ca^2+^-transporter, and, amongst others, *fei1* and *fei2* single mutants [49]. Intriguingly, *fei2* mutants showed suppressed responses to isoxaben similar to *mca1* and the *the1-1* loss-of-function allele. This suppression was more pronounced in *fei2 the1-1* double mutants than in *fei2* single mutants, but was identical to *the1-1*. By contrast, the abnormally increased *THE1* activity in the hypermorphic *the1-4* allele showed elevated responses to isoxaben, while the isoxaben responses in the *fei2 the1-4* double mutant largely resembled the wild type. This suggested a linear genetic interaction between *THE1, FEI2,* and *MCA1* in CWD response, where *THE1* was proposed to act upstream of *FEI2* and *MCA1* [49]. However, I am viewing the latter interpretation with reservation. On the one hand the hypermorphic *the1-4* allele showed a significant effect on CWD response in both the *fei2* and the *mca1* background, which is inconsistent with a strictly linear model with *FEI2* and *MCA1* nonredundantly acting downstream of *THE1*. On the other hand, *FEI2* is known to be partially redundant with FEI1, which might explain its reduced effect compared with *THE1*. Since *fei1 fei2* double mutants were not tested for their hormonal CWD response, it is possible that the *FEI1* locus partially fulfils the same function as FEI2, not only in vegetative growth, but also in this response and that *THE1* acts on both of the RLKs. This would explain why a hypermorphic *the1-4* allele can still act in a *fei2* background, even when acting upstream of the *FEI* loci. In summary, the nonadditive genetic interaction between *THE1* and *FEI2* in the response to CWD suggests that *FEI2* is at least partially required for *THE1* action in CWI signaling and vice versa.

A possible controversy between observations related to the role of the FLA4-FEI pathway to CWD response is the hypersensitivity to isoxaben of *fei1 fei2* with respect to root growth [27,43] on the one hand and the repression of the hormonal response to isoxaben *fei2* single mutant on the other hand [49]. However, the apparent hypersensitivity of *fei1 fei2* to isoxaben observed in the root growth assay might in fact be due to a lack of an appropriate response to CWD similar to what was recently observed in *fer* mutants transferred to high salt medium [50].

Interestingly, the study by [49] not only revealed the synergistic interaction between *THE1* and *FEI2* in the CWD response. These two loci were also shown to antagonize pattern-triggered immunity (PTI) mediated by the genetic loci encoding signaling peptides PEP1 and PEP3 and their receptors PEPR1 and PEPR2. This suggested that the FLA4-FEI pathway might play a role in the balance between growth and PTI. However, whether *FEI1* acts in partial redundancy with *FEI2* and whether or not *FLA4* acts in the CWD response pathway are open questions.

Besides the activation of JA and SA signaling, another feature of the response to cellulose synthesis inhibition was a biphasic increase of reactive oxygen species (ROS) production that depended on *RESPIRATORY BURST OXIDASE HOMOLOG D (RBOHD)* [51]. The *fla4* mutant exposed to elevated salt levels also showed increased production of ROS and ectopic lignification [52]. In contrast to cellulose biosynthesis inhibition, the moderately increased ROS production in *fla4* did not depend on *RBOHD* or its paralog *RBOHF*. This means that the *FLA4-FEI* pathway might act in the control of ROS in more than one way: as a negative regulator of *RBOHD/F*-independent ROS production in root growth and as a positive regulator of *RBOHD*-dependent ROS production triggered by the CWD response pathway. To summarize, in this section I discussed the genetic analyses that place the *FLA4-FEI* pathway in the role of regulating both CWI as well as CWD responses, and its interaction with PTI and ROS level.

## 4. Interactions with Growth Regulator Signaling

### 4.1. Control of Ethylene-Independent ACC Signaling

Apart from its involvement in CWI and CWD response, several recent studies have studied the interactions of the FLA4-FEI pathway with small-molecule growth regulators. The nonproteinogenic amino acid, 1-aminocyclopropane 1-carboxylic acid (ACC), is the direct precursor of ethylene and in this role is crucial for the control of both abiotic and biotic stress, as well as fruit-ripening and senescence. However, it has recently become clear that ACC must have important functions independent of ethylene signaling [reviewed by [53]. One of the first ethylene-independent ACC responses to be described was the suppression of the *fei1 fei2* phenotype by the drug α-aminobutyric acid (AIB) that blocks the conversion of ACC to ethylene and it was suggested that AIB might block a putative ACC-receptor, whose overactivation in the *fei1 fei2* mutant might cause the mutant phenotype [27]. This suggestion, in combination with the physical interaction between the FEIs and some ACS isoforms, led to the proposal that the FEIs might control ACC biosynthesis close to the plasma membrane.

Coming from a different angle, Nühse and coworkers pharmaco-genetically studied the effect of isoxaben on cell expansion [54]. The length of the first fully elongated root hair-forming cell in the growing root (LEH) was significantly reduced already 0.5 h after the application of ACC. This effect on cell elongation was blocked by the ACS inhibitor aminoethoxyvinylgylcine (AVG) and other ACS inhibitors, but not by silver ions that block ethylene perception, demonstrating that ACC mediated the short-term response of elongation growth to cellulose biosynthesis inhibition in an ethylene-independent fashion. Roots treated with the combination of isoxaben and AVG continued to expand but took on abnormal shapes, probably due to the biomechanical role of cellulose, while root epidermal cells treated with isoxaben alone expanded less but they kept a normal shape. Interestingly, after long-term exposure to ACC, the inhibitory effect on root elongation became dependent on ethylene signaling, suggesting that the ACC-specific effect on root epidermal cell expansion was transient. Using additional inhibitors and mutant backgrounds, they also implicated auxin signaling and ROS production in the ACC-mediated reduction of root elongation that was triggered by cellulose synthesis inhibition [54]. The ethylene-independent repression of cell expansion by ACC was later also observed in non-root tissues [55].

Due to ‘masking’ by ethylene signaling, relatively little is presently known about the mechanism of ACC-specific signaling. Interestingly, it has been shown that ACC induces Ca^2+^-currents in ovules via GLUTAMATE RECEPTOR HOMOLOGS (GLRs) mediating the release of pollen tube chemoattractant [56]. The sufficiency of ACC to trigger GLR-dependent ion fluxes was shown in transfected mammalian cells, thus demonstrating its direct and ethylene-independent mode of action. In addition, ACC was significantly more potent in this assay than any other tested amino acid, including glutamate. ACC sensing by GLRs and subsequent Ca^2+^-based signal transduction offers a possible mode of action for ACC in CWI signaling as a downstream process of the FLA4-FEI pathway.

The kinase domains of the FEIs interacted with the type 2 isoforms ACS5 and ACS9 but not the type 1 ACS2 [27] (for ACS typology see [57]). What is so far unknown is if and how the FEIs regulate ACS in vivo. However, based on genetic and pharmacological evidence, one could speculate that in a resting state the FEIs repress ACS, thereby allowing unhindered cell elongation. This resting state might theoretically involve direct binding of FLA4 to the FEI extracellular domain [58] or to a FEI-containing receptor complex. Therefore a lack of either component would lead to constitutive activation of ACS and to repression of cell elongation. This possibility seemingly contradicts the suppression of the *fei1 fei2* mutant phenotype by AIB. AIB is known to block ACO and might lead not to a reduction of ACC, but rather to an accumulation, hence it was suggested that AIB, which is structurally analogous to ACC, might suppress the *fei1 fei2* phenotype by competitively blocking a hypothetical ACC-receptor [27]. It will be interesting to test if AIB suppresses the recently demonstrated stimulatory effect of ACC on GLRs.

How could ACC affect the cell wall? A direct activation of Ca^2+^-signaling by ACC can rapidly affect many processes, including short-term salt responses [59], ROS signaling [60], and vacuolar biogenesis [61]. On the transcriptional level, ACC has been shown to affect the expression of genes such as multiple peroxidases and extensins that act in cell wall crosslinking in an ethylene-independent fashion [62]. So far, the effect of the FLA4-FEI pathway on the transcriptome has not been described, however, it would not be surprising to find ACC-responsive transcripts among FLA4-FEI-repressed genes. Potentially, ACC might also have ethylene-independent effects on cell wall formation, e.g., by influencing cellulose synthase [63]. In summary FLA4-FEI-repressed ACS could allow unimpeded cellulose biosynthesis and cell stretching in an optimally growing plant and in challenging conditions, such as pathogen attack or abiotic stress, ACC would throttle cellulose biosynthesis and, via transcriptional activation, increase the relative level of oxidative cell wall crosslinking.

### 4.2. Interactions with Abscisic Acid (ABA) and Auxin Signaling

Apart from its presumed role in the control of ACC, genetic and physiological studies have previously implicated the FLA4-FEI pathway with auxin and abscisic acid (ABA). The aforementioned suppressor screen in the *fei1 fei2* background also identified *shou2* mutant alleles of the *IAA-alanine resistant 4* (*IAR4*) locus and found that *iar4* and mutant alleles of other IAA biosynthetic loci (*WEI8* and *TAR2*) partially suppressed *fei1 fei2* and *fla4* and several other cell wall-defective mutants [63]. This finding suggested a role for auxin in the control of CWI—possibly by activating ACS in antagonism to the FLA4-FEI pathway.

ABA is the most important small molecule regulator of the genomic responses to abiotic stress [64]. When the response of *fla4* mutants to various growth regulators and related substances was tested, it was found that ABA suppressed the *fla4* mutant root phenotype. Consistently, *fla4* was also suppressed by ABA-oversensitive mutants *cpl1* and *sad1.* Conversely, chemical inhibition of ABA biosynthesis in wild type plants phenocopied the salt-oversensitive *fla4* phenotype [65]. Intriguingly, also NaCl-triggered closure of stomata in leaves strongly depended on *FLA4* and was partially restored by application of ABA to *fla4* mutants. Moreover, the ABA content in NaCl-treated seedlings was significantly lower in *fla4* mutants compared to the wild type [66], a trend which followed the transcript level of ABA metabolic loci showing upregulation for ABA catabolic enzymes and downregulation for ABA biosynthetic loci [65]. These observations suggested a synergistic but largely independent action of the *FLA4-FEI* pathway and ABA on abiotic stress responses and cell expansion, which is consistent with the unexpected role for the *FLA4-FEI* pathway in the control of endogenous ABA. An alteration in ABA level might explain some abiotic stress-related aspects of the *fla4* and the *fei1 fei2* phenotype.

ABA is known to maintain turgor pressure during salt and drought stress by regulating the biosynthesis of osmolytes [67] and also controls ROS in both a positive and negative fashion [68,69]. It is presently unknown how *FLA4-FEI* signaling regulates ABA metabolism. One possibility is that ethylene negatively regulates ABA biosynthesis like previously observed [70]. Generally, ethylene has been shown to interact with most other major stress response pathways including jasmonate and salicylate signaling [71]. However, it is also possible that ACC plays an ethylene-independent role in the interaction with other stress response pathways and a careful genetic dissection discriminating direct and indirect effects of ACC and ethylene in their cross-talk to growth-regulator signaling remains to be undertaken. In summary, genetic and physiological studies have identified several loci that act in a linear genetic interaction with one or more components of the FLA4-FEI pathway or act in synergism with it. A network of possible interactions based on isolated previous observations is shown in Figure 1.

## 5. The FLA4-FEI Pathway Seen from the Transcriptomic Perspective

Is it possible to speculate about mechanisms, based on genetic evidence alone? Might some of the proteins encoded by genetically interacting loci also interact physically? Sometimes it can be observed that loci interacting in the same biochemical pathway are co-regulated at the transcript level [72,73]. In turn, transcriptional co-expression can serve as a hypothetical lead for the discovery of new components of a genetic pathway or a biochemical process [74,75,76].

While *FLA4* is not represented on the most popular *Arabidopsis* oligonucleotide chips, its expression pattern in many different tissues and developmental stages is covered in a comprehensive gene expression atlas based on RNA sequencing [77]. From the publicly available RNA sequencing data, pairwise correlation coefficients (Pearson′s) were generated (kindly provided by Anna Klepikova). Table 1 lists the previously discussed genetic ‘interactors’ of the *FLA4-FEI* pathway sorted by their degree of co-expression with *FLA4*. From this table it can be seen that, as expected from their organism-wide non-additive genetic interaction, the ‘core components’ of the *FLA4-FEI* pathway, i.e., *FLA4*, *FEI1,* and *FEI2,* were co-expressed from a moderate to high degree. Likewise, *THE1,* which acts in a linear genetic pathway with *FEI2* in response to cell wall biosynthesis inhibition [49], showed a moderate to high co-expression with all three core components (Table 1). By contrast, most of the loci that acted independently of the *FLA4-FEI* pathway and also the seed coat-acting *FLY1* and *MUM2* loci, were not co-expressed with FLA4 or the FEIs. Surprisingly, two loci acting in cellulose biosynthesis in the seed coat, *CesA5* and *COBL2*, that were previously found to act in a role independent from *FLA4* and *FEI2* [44,45,46], showed a relatively high degree of co-expression with the *FLA4-FEI* core components. In fact, the *CesA1*, *CesA3*, and *CesA6* loci acting in primary cell wall biosynthesis showed a high degree of co-expression with FEI1 and at least moderately high co-expression with the other two core components, while the secondary cell wall active *CesA4, -7 and -8* were less co-expressed, which supports the suggestion that the *FLA4-FEI* pathway predominantly affects CWI in the primary cell wall, where it might tightly interact with *THE1* (Table 1). In summary, the role of the FLA4-FEI pathway in primary cell wall biogenesis and in the THE1-dependent CWD response is confirmed at the transcriptome level.

## 6. The Relevance of FLA4-FEI Pathway for Agronomic Traits

Molecular genetic studies in model organisms are valuable for fundamental research, however, they don′t always represent evolutionary processes in wild crop ancestors and crop cultivars, where multiple co-evolving loci quantitatively contribute to different traits. However, given its multiple connections to growth control and stress response in *A. thaliana*, the *FLA4*-*FEI1* pathway might regulate traits of agronomic importance in crop plants. Among numerous genetic loci implicated with the phenomenon of leaf rolling, a putative rye (*Secale cereale)* orthologue of the *FEIs* showed an association to this trait [80]. Leaf rolling occurs in grasses as an adaptive response to drought. A role of *FEI* in leaf rolling is thus consistent with a conserved role of the *FLA4-FEI* pathway in both abiotic stress and CWI.

Another discovery linking the *FLA4-FEI* pathway to a trait of agronomic importance revealed that the aggressive fungal pathogen *Botrytis cinerea* produces small RNAs (Bc-sRNAs) that interfere with the expression of genes involved in plant immunity [81]. One of these Bc-sRNAs, Bc-siR37, suppressed transcripts of *FEI2* in *A. thaliana* and, interestingly, *fei2* mutant leaves showed enhanced susceptibility to *B. cinerea* [82]. Remarkably, also *THE1* was previously shown to be required for normal resistance to *B. cinerea* [83] and, as explained above, *FEI2* was suggested to act in a linear fashion in the pathway together with *THE1* in CWD response, counteracting PTI [49]. This suggests that *B. cinerea* has identified *FEI2* as an Achilles heel in the innate defense mechanisms of host plants, possibly because of the FLA4-FEI pathway′s role in balancing growth with stress-responses.

## 7. Too Many FEI Interactors! Which Ones Are Relevant?

So far, I have discussed that the *FLA4-FEI* pathway is involved in various physiological processes that were analyzed with genetic and pharmacological tools. From these multiple studies, we get a very complex and rather diffuse picture of the regulatory network (Figure 1). The core components, with the potential inclusion of THE1 and MCA1 on the same level as the FEIs and with ACS as downstream effector, present the backbone of the pathway that interacts with multiple other processes related to primary cell wall function and growth regulator signaling (Figure 1). What is the mechanistic basis of these genetic interactions?

Apart from the isolated observation of yeast two-hybrid binding between the FEIs and ACS5 and ACS9, no physical interactions have been characterized in detail [27]. While fluorescent protein-tagged versions of FEI1 and FLA4 have been shown to localize to the plasma membrane in separate studies [27,48], and structural modeling illustrated how FLA4 might theoretically bind to FEI1 [58], there is no experimental proof of the co-localization of FLA4 and the FEIs or their direct interaction. The mining of protein–protein interaction (PPI) datasets together with transcriptional co-expression might shed further light on the mechanistic basis of the FLA4-FEI pathway.

A large-scale PPI study focusing on the LRR-RLK superfamily expressed the extracellular domains (ECDs) of most of the 236 *A. thaliana* LRR-RLKs in insect cells as secreted epitope-tagged proteins and tested pairwise co-immunoprecipitation [79]. The FEIs were involved in three of the identified 567 high-confidence interactions (HCIs) (Table 2). The two identified FEI interactors, AT2G02780 and AT1G17230, are both previously uncharacterized LRR-RLKs and were altogether involved in twenty-eight (incl. FEI1 and FEI2) and ten (incl. FEI1) HCIs, respectively. At the transcript level, *AT2G02780* showed moderate co-expression with both FEIs and with FLA4, while *AT1G17230* was moderately co-expressed with FEI1 (Table 2).

An even broader PPI screen included many different membrane proteins potentially involved in trafficking and signaling, and utilized the split-ubiquitin yeast two-hybrid technology. This study identified more than 10,000 interactions among 6.4 × 10^6^ tested pairwise combinations [84]. Both FEIs were found to interact with several other proteins (Table 2). The identified interactors were LRR-RLKs and other predicted receptor-like proteins, but also proteins involved in membrane trafficking, ion transporters, enzymes, and numerous proteins of unknown function. However, proteins predicted to be localized in mitochondria or chloroplasts that might either be wrongly annotated or false positives were identified as well. Among the many potential FEI interactors [84], it is worth mentioning the *EMB3135* locus [86] that encodes a novel predicted plasma membrane protein [85] and is highly co-expressed with *FLA4* and *FEI2* (Table 2). In the split ubiquitin assay, EMB3135 interacted with almost 200 other bait proteins [84], however, the essentiality for embryo development aside, its function remains to be investigated.

As a guiding principle for LRR-RLK function [79], it was postulated that LRR-RLKs with large ECDs (>12 LRRs) are involved in signal perception, while RLKs with smaller ECDs act as co-receptors or in scaffolding and thereby modulate perception. FEI1 and FEI2 have five LRRs and on that basis were predicted to be co-receptors [78]. A co-receptor or scaffolding role of the FEIs is consistent with the observation that protein kinase activity was not essential for the genetic function of recombinant *FEI1*, but might have contributed to the robustness of its function [27]. Given the many candidates for PPI partners, it is possible that depending on the tissue or stress condition, the FEIs could cluster with different receptors, such as THE1 and other LRR-RLKs that might then act as sensors. Briefly, the FEIs might be regulators in one or more larger sensor complexes rather than being sensors in their own right.

There is even less clarity what FLA4′s role in such a sensory complex might be. Like most FLAs, FLA4 is anchored to the outer sheet of the plasma membrane by a glycosylphosphatidylinositol (GPI) anchor. Functional fluorescent protein-tagged FLA4 localized to the plasma membrane and was partially released to the apoplast of roots [48] and developing seed coat epidermal cells [45]. Crucially, when the C-terminal sequence of FLA4, which is essential for GPI anchor attachment, was deleted, the construct encoding the resulting soluble FLA4 complemented the *fla4* mutant [45,48]. This finding suggested that FLA4 might function as a soluble signal. Despite FLA4 and its angiosperm relatives always consisting of two Fas1 domains, constructs with deletions of the amino-proximal Fas1 domain did not only complement the *fla4* mutant, but even had a promoting effect on root elongation under some growth conditions [48,58]. A structural model based on the assumption that FLA4 and FEI1 physically interact showed that FLA4′s essential carboxy-terminal Fas1 domain had interaction sites with FEI1 and that one of the predicted interaction hotspots was the amino acid residue that was substituted in the initially described *FLA4* mutant allele *sos5-1* [58]. Interestingly, the model suggested considerable flexibility around the central proline-rich hinge domain connecting the Fas1 domains, allowing a competition between the intermolecular interaction of FLA4 with FEI1 and the intramolecular interaction between the two Fas1 domains of FLA4. This was reminiscent of a previous suggestion that human Fas1 proteins binding to cell surface integrins might self-regulate this binding by intramolecular competition [87]. Thus, while interaction between the carboxy-proximal Fas1 domain of FLA4 with the FEIs might be crucial to the overall function of the pathway, the intramolecular interaction between the two FLA4 Fas1 domains might fine-tune its activation state.

Seemingly, the hardest problem in plant CWI signaling is the molecular mode of interaction between the proteinaceous components of the signaling pathway and the cell wall itself, and the FLA4-FEI pathway is no exception. The only thing that is presently known is that FLA4 is partially secreted to the apoplast and can, in principle, act as a soluble molecule, but it is not known if and how the chemical and biomechanical environment in the apoplast might influence its activity. So far, only two examples for receptor-like proteins physically interacting with cell wall polysaccharides exist. Firstly, WAKs bind to the cell wall in a pectin-dependent fashion in vivo and they directly and noncovalently bind to pectic fragments in vitro (reviewed in [88]). Secondly, the malectin-like domain RLK FER was found to interact with pectic homogalacturonan in vitro [50]. However, this finding has, until now, not been reproduced for any other malectin-like domain RLK. In fact, based on crystal structures, molecular modeling, and the negative results of isothermal titration calorimetry experiments, it was suggested that the carbohydrate-binding surface of the malectin-like domain might have evolved into a PPI domain in plants and might not actually bind to carbohydrates [89,90]. Consequently, FER was found to bind to several proteins including multiple LRR-extensin (LRX) proteins, which might link FER to the cell wall in a fashion that is still not entirely clear [91,92]. However, as far as components of the FLA4-FEI pathway are concerned there is presently no indication of direct or indirect interactions with cell wall polymers.

## 8. Conclusions

Combining aspects of previous hypotheses for modes of action of the FEIs and for FLA4, I envisage that FEIs might be important co-receptors in larger receptor complexes, the precise composition of which might vary from tissue to tissue. One of the interacting receptors might be *THE1,* but other LRR-RLKs and transporters, such as *MCA1,* might also be involved. FLA4 might act as a regulatory ligand interacting with the hypothetical complex via its carboxy-terminal domain and could autoregulate the degree of this interaction by a jackknife-like action of the amino-terminal Fas1 domain, which might be under the influence of the extracellular biochemical and biomechanical environment. A downstream client of the module might be ACS. The resulting ACC might fulfil multiple effects on growth regulators, ROS, and cell wall crosslinking, both directly by its potential to regulate intracellular Ca^2+^ levels via GLR activation or indirectly by the regulation of ACC-responsive loci. The most uncertain field, however is the mode of interaction of the plasma membrane with the cell wall and a close interaction between the FLA4-FEI pathway to pectin is only supported by genetics. Due to its many interactions to cell wall structure and cellular signaling, the FLA4-FEI module might participate in the regulation of multiple agronomic traits involving the balance between growth and stress-response.

## Figures and Tables

**Figure 1 genes-12-00145-f001:**
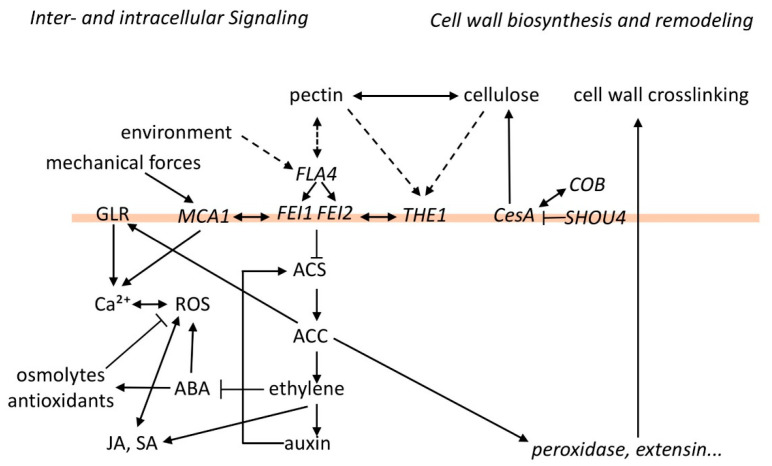
Hypothetical relations between cell wall polymers and stress signaling mediated by the *FLA4-FEI* pathway. At the center of the pathway *FLA4* and the *FEIs* might repress 1-AMINOCYCLOPROPANE 1-CARBOXYLIC ACID SYNTHASE (ACS), which could affect various other growth regulators, including abscisic acid (ABA), auxin, ethylene, jasmonic acid, and salicylic acid. ABA regulates osmotic and oxidative stress responses. ACC influences intracellular calcium by activating GLUTAMATE RECEPTOR HOMOLOGS (GLRs) and also affects transcription of cell wall peroxidases and extensins, which could crosslink cell wall polysaccharides. There appears to be a tight interaction between the *FLA4-FEI* pathway and *THE1* as well as *MCA1* in cell wall integrity (CWI) sensing. The physical interactions that connect cell wall polymers with the *FLA4-FEI* pathway are presently unknown (dashed arrows).

**Table 1 genes-12-00145-t001:** Genetic interactors of the FLA4-FEI pathway.

AGI	Name	Molecular Function *^)^	Type of Interaction ^x)^	Localization ^@)^	Co-Expression ^§)^ R^2^		
					FLA4	FEI1	FEI2
AT3G46550	FLA4	unknown	D + [27]	PM	1	0.58	0.70
AT2G35620	FEI2	co-receptor	D + [27]	PM	0.70	0.42	1
AT5G54380	THE1	RLK	D + [49]	PM	0.59	0.55	0.33
AT1G31420	FEI1	co-receptor	D + [27]	PM	0.58	1	0.42
AT5G64900	PROPEP1	PTI peptide	I − [49]	vacuole	0.52	0.02	0.59
AT5G53340	HPGT1 ^1)^	AGP biosynt.	n.d.	med-Golgi	0.46	0.17	0.37
AT5G09870	CESA5	seed coat secondary cell wall cellulose	I + [46]	PM	0.46	0.81	0.34
AT1G73080	PEPR1	receptor	I − [49]	PM	0.44	−0.02	0.45
AT4G39350	CesA2 ^1)^	seed coat secondary cell wall cellulose	n.d.	PM	0.44	0.57	0.63
AT4G21060	GALT2	AGP biosynt.	D + [47]	Golgi	0.42	0.09	0.35
AT2G25300	HPGT3 ^1)^	AGP biosynt.	n.d.	Golgi	0.41	−0.05	0.35
AT4G32410	CesA1 ^1)^	primary cell wall cellulose	n.d.	PM	0.41	0.75	0.25
AT5G05170	CesA3 ^1)^	primary cell wall cellulose	n.d.	PM	0.37	0.69	0.36
AT5G64740	CesA6	primary cell wall cellulose	I + [27]	PM	0.37	0.68	0.23
AT3G29810	COBL2	seed coat secondary cell wall cellulose	I + [46]	PM	0.34	−0.05	0.57
AT1G17750	PEPR2	receptor	I − [49]	PM	0.32	0.16	0.19
AT4G22290	SHOU4L2	trafficking	I − [43]	PM	0.24	0.10	0.57
AT4G35920	MCA1	Ca^2+^ transport	D + [49]	PM	0.24	0.17	0.17
AT5G60920	COB	primary cell wall cellulose	I + [27]	PM	0.23	0.68	0.11
AT4G32120	HPGT2 ^1)^	AGP biosynt.	n.d.	Golgi	0.15	-0.07	0.31
At1g70560	WEI8	auxin level	I − [63]	cytosol	0.08	−0.25	−0.05
AT4G18780	CesA8 ^1)^	secondary cell wall cellulose	n.d.	PM	0.08	0.35	0.22
At4g24670	TAR2	auxin level	I − [63]	vac.	0.07	−0.33	0.28
AT5G17420	CesA7 ^1)^	secondary cell wall cellulose	n.d.	PM	0.07	0.33	0.22
AT5G44030	CesA4 ^1)^	secondary cell wall cellulose	n.d.	PM	0.05	0.32	0.20
AT2G21770	CesA9 ^1)^	seed coat secondary cell wall cellulose	n.d.	PM	0.05	−0.13	0.01
AT1G78880	SHOU4	trafficking	I − [43]	PM	0.01	−0.21	0.09
AT5G64905	PROPEP3	PTI peptide	I − [49]	vac.	0.00	−0.09	0.12
AT1G74800	GALT5	AGP biosynt.	D + [47]	Golgi	−0.02	−0.13	0.00
AT5G48870	SAD1	ABA signaling (neg. regulator)	I − [65]	cytosol, nuc.	−0.16	−0.39	0.01
AT4G21670	CPL1	ABA signaling (neg. regulator)	I − [65]	nuc.	−0.22	-0.26	−0.31
AT1G24180	IAR4	auxin level	I − [63]	mitoch.	−0.23	−0.26	−0.33
AT2G40220	ABI4	ABA signaling	I + [65]	nuc.	−0.28	−0.20	−0.20
AT2G36270	ABI5	ABA signaling	I + [65]	nuc.	−0.29	−0.3	−0.35
AT5G63800	MUM2	pectin in seed coat mucilage	D − [44]	apopl.	−0.31	0.17	−0.39
AT4G28370	FLY1	pectin in seed coat mucilage	D − [44]	unkown/novel compartment	−0.35	−0.15	−0.16

*^)^ Predicted molecular function is from TAIR (arabidopsis.org). LRR-RLKs were further classified into receptors and co-receptors according to [78]. ^@)^ Predicted localization is from SUBA (https://suba.live) [79]. ^§)^ Co-expression values R^2^ (Pearson′s correlation coefficient) were calculated from expression values obtained from TRAVA (http://travadb.org) [77] and kindly provided by Anna Klepikova (The Institute for Information Transmission Problems, Moscow, Russian Federation). For quick recognition R^2^ values ≥0.5 are in red, R^2^ values ≥0.1 and <0.5 are in green, and R^2^ values <0.1 are in blue. ^x)^ The indicated type of genetic interaction are my interpretations of published genetic experiments (double and higher order mutants): D: genetic action depends on FLA4-FEI, I: genetic action is independent from FLA4-FEI, +: synergistic action, −: antagonistic action, n.d.: (direction of) interaction has not been ascertained. ^1)^ Due to their possible relevance for the pathway HPGT1, -2 and -3 as well as CESA1,-2,-3, -4, -7 -8 and -9 are also included in this list, although genetic interactions with FLA4-FEI core components have not been published.

**Table 2 genes-12-00145-t002:** Physical interactors of the FLA4-FEI pathway.

AGI	Name	Molecular Function *^)^	Partner in Pathway	Expected Localization ^@)^	Co-Expression ^§)^ R^2^		
					FLA4	FEI1	FLA2
AT5G11890	EMB3135	unknown	FEI1, FEI2 [84]	PM	0.71	0.35	0.61
AT2G02780	LRR IV	co-receptor	FEI1, FEI2 [83]	PM	0.49	0.23	0.30
AT1G34470	unknown	unknown	FEI1 [84]	mt, cp	0.38	0.49	0.36
AT2G40316	autophagy-like	unknown	FEI1 [84]	vac.	0.36	0.21	0.419
AT2G29180	unknown	unknown	FEI1 [84]	cp	0.28	0.551	0.179
AT5G59650	LRR I	receptor	FEI2 [84]	PM	0.20	0.207	0.307
AT5G49540	Rab5-interacting fam.	trafficking	FEI1, FEI2 [84]	PM	0.17	−0.1	0.272
AT5G06320	NHL3	unknown	FEI2 [84]	PM	0.145	0.112	−0.1
AT3G28220	TRAF-like fam.	unknown	FEI1 [84]	cp	0.074	−0.01	-0.2
AT1G47640	seven transmembrane domain fam.	receptor-like	FEI1, FEI2 [84]	PM, apopl.	0.07	−0.17	0.19
AT2G31360	acyl-CoA desaturase like	enzyme	FEI1 [84]	ER	0.06	−0.06	0.42
AT5G47530	auxin-responsive fam.	unknown	FEI1 [84]	PM, apopl.	0.05	0.16	0.39
AT5G14030	translocon-β	unknown	FEI1 [84]	ER	0.01	−0.25	0.087
AT3G12180	cornichon fam.	trafficking	FEI2 [84]	apopl.	0.00	−0.21	0.13
AT4G25360	TBL18	O-acetyl transferase	FEI2 [84]	mt	0.00	0.07	−0.31
AT4G14455	BET12	trafficking	FEI1, FEI2 [84]	Golgi	0.00	−0.1	0.08
AT1G70520	CRK2	RLK	FEI1 [84]	PM	−0.03	0.25	−0.20
AT4G05370	unknown	unknown	FEI1, FEI2 [84]	apopl., PM	−0.07	−0.17	−0.2
AT5G65800	ACS5	ACC biosynt.	FEI1, FEI2 [27]	cytosol, PM	−0.10	0.02	−0.03
AT4G20790	LRR VI	co-receptor	FEI2 [84]	PM	−0.10	−0.18	−0.18
AT3G49700	ACS9	ACC biosyn.	FEI1, FEI2 [27]	cytosol, PM	−0.11	−0.02	−0.02
AT5G47180	VAMP fam.	trafficking	FEI2 [84]	ER	−0.16	−0.15	−0.26
AT2G26180	IQD6	microtubule organization	FEI1, FEI2 [84]	nuc., cytosol	−0.18	-0.29	0.15
AT5G40640	unknown	unknown	FEI1 [84]	PM	−0.18	−0.37	−0.11
AT1G33100	MATE efflux fam.	ion transport	FEI1 [84]	PM, vac.	−0.19	−0.13	−0.06
AT4G23220	CRK14	RLK	FEI1 [84]	PM	−0.20	−0.02	−0.26
AT1G17230	LRR XI	receptor	FEI1 [79]	PM	−0.20	0.24	−0.18
AT1G21240	WAK3	RLK	FEI1, FEI2 [84]	PM	−0.23	−0.07	−0.29
AT4G37680	HHP4	receptor-like	FEI1, FEI2 [84]	Golgi, vac., PM	−0.24	−0.19	−0.36
AT2G41705	CrcB fam.	fluoride transporter	FEI1, FEI2 [84]	PM	−0.30	−0.23	−0.31
AT5G27350	major facilitator fam.	sugar transporter	FEI1 [84]	PM, vac.	−0.30	−0.08	−0.35
AT3G10640	VPS60.1	trafficking	FEI2 [84]	nuc.	−0.31	−0.21	−0.44
AT1G17280	UBC34	ubiquitination	FEI1, FEI2 [84]	perox.,cytosol,	−0.34	−0.19	−0.48
AT3G17000	UBC32	ubiquitination	FEI1, FEI2 [84]	cytosol	−0.35	−0.29	−0.47
AT1G29060	SFT12	QcSNARE trafficking	FEI2 [84]	Golgi	−0.39	−0.25	−0.40
AT2G04040	MATE efflux fam.	ion transport	FEI1 [84]	PM	−0.40	−0.30	−0.39
AT4G04860	DER2.2	ubiquitination	FEI1 [84]	ER, PM	−0.41	−0.35	−0.39
AT3G66654	CYP21-4	ABA signaling	FEI1 [84]	Golgi	−0.5	−0.13	−0.56

*^)^ Predicted molecular function is from TAIR. LRR-RLKs were further classified into receptors and co-receptors according to [78]. ^@)^ Predicted localization is from SUBA [85], except when experimental studies showed different localization. ^§)^ Co-expression values (Pearson′s correlation coefficient) were calculated from expression values obtained from TRAVA db [77]. Coloring of R^2^ values see Table 1.

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
