# Peer review of "The FLA4-FEI Pathway: A Unique and Mysterious Signaling Module Related to Cell Wall Structure and Stress Signaling"

_genes, 2021, doi:10.3390/genes12020145_

Round 1
Reviewer 1 Report
This is a comprehensive review of the function of the plant protein complex of FASCICLIN LIKE ARABINOGALACTAN PROTEIN 4 (FLA4) and two leucine-rich repeat receptor kinases, FEI1 and FEI2. I did find the review to be challenging to read at times with one to two page length paragraphs that describe the details, mostly genetic interactions. I do think the audience, especially those only immediately outside the specific area reviewed, would benefit from a more basic introduction to each topic as well as concluding remarks.
For example, some of the paragraphs lack structure and I offer the example of the paragraph that begins on line 128 and ends line 172. You will note that the first sentence does not introduce a clear topic of the paragraph nor does the last sentence conclude it in any way. Intro sentence - When the combined phenotypic evidence gathered from fla4 and fei2 single and fei1 fei2 double mutants is considered, the observations are comparable with mutants in pectin biosynthesis such as abnormally thin middle lamella and disrupted cell adhesion. Conclusion sentence - This interpretation is consistent with the demonstration that an engineered FLA4 variant lacking the putative O- glycosylation sites remained genetically functional (Xue et al. 2017). I do not think it is necessary to compromise clarity with such a comprehensive review. Other paragraphs to consider for improved paragraph structure begin lines 128 and 220.
All of the information appears to be included in the review, but it could be made more accessible in shorter paragraphs with more structure. I also note some typos below.
Line 20. The very first sentence of the introduction is not an introduction to the paragraph or the review. I suggest it be removed.
Line 22. “…that do not exist…”
Line 25. “plant live” does not make sense. If this is meant to be life I would challenge the author and suggest that ALL aspects of plants ate crucially impacted by cell wall integrity. This, I find this an over statement.
Line 55. Replace “The knowledge” with “Indetification”
Line 57. Should use A. thaliana after the first mention of Arabidopsis thaliana.
Line 117. I disagree with this statement. I can’t think of a single model of the cell wall that does not involve interactions of these three groups of polymers. “While cellulose and hemicellulose on the one hand and pectin on the other hand have long been regarded as forming largely independent networks in the cell wall,….” They may exist, but I think it is well accepted that the wall polymers exist in a network, or matrix.
Line 203. This deserves to be more than one sentence. “Interestingly, the study by (Engelsdorf et al. 2018) also revealed an antagonisticinteraction between CWI signaling by THE1 and FEI2 on the one hand and pattern-triggered immunity (PTI) mediated by the genetic loci encoding signaling peptides PEP1 and PEP3 and their receptors PEPR1 and PEPR2 on the other hand, thereby providing a potential signaling module thatcould act in the balance between growth and PTI.”
Line 215. …in more than one way
Line 221. …control
Line 250 …than any other…
Figure 1 caption typos. Cell should be lowercase. FLA4 and and FEIs
Line 334 …from a moderate…
Table 1. The different colors for the values need to be defined.
Line 398. At the transcript level…
Line 408. Among the many…
Line 406. However, proteins predicted….
Line 408. Among the many….
Line 446. …if and how…
Line 459 …pectin-dependent fashion…
Author Response
I am grateful for the reviewer's constructive criticism. My reply is found in the attached document.

Reviewer 2 Report
Plant cell wall integrity sensing is a signaling process by which plants sense the correct deposition and integrity of their cell wall extracellular matrices. While this process is mechanistically poorly understood, various signaling factors have been implicated in genetic screens and genetic interaction studies. The FEI receptor like kinases and the Fasciclin-Like Arabinogalactan protein 4 (FLA4) protein are among these previously identified regulatory components, and genetic studies suggest that these genes participate in a signaling pathway together. However, additional studies have implicated a wide variety of additional factors that impinge on the FEI-FLA4 signaling pathway.
The author of this review provides a very nice summary of what is currently known about the FEI-FLA4 signaling pathway. He additionally discusses the interactions of this pathway with various hormonal and abiotic stress signaling pathways, provides a current model of how this pathway may participate in cell wall integrity sensing, and discusses future directions that need to be addressed to gain stronger mechanistic insight into the function of these proteins.
This is a nice review that summarizes and discusses a relatively understudied area of plant cell wall integrity sensing. I am not aware of a previously published review that focuses on this topic, the author has done a good job of considering the area as a whole, and I believe that this manuscript may be highly cited because of its unique focus.
My only comment on the manuscript is that there are a number of very minor grammatical errors distributed throughout the manuscript (a few of which I have highlighted below), and it would be good for the author to simply check over the manuscript an additional time to make sure that these very small mistakes are corrected.
- Line 25: Change “plant live” to “plant life”
- Line 26: Change to “has been genetically dissected”
- Line 71: Remove “is and” before “acts”
- Line 81: Change “leucin” to “leucine”
- Line 141: No comma in the following sentence “for seed coat adhesion, suggested…”
- Line 145: Fix “@” symbol to “α”
- Line 225: Fix “@” symbol to “α”
Author Response
I am grateful to the reviewers's encouraging comments. My specific responses are described in the attached document.

Reviewer 3 Report
Author has pointed out rightly that an in planta activation of FEI-FLA4 signaling pathway will be very insigtfull. One option will be to employ genetically encoded biosensors like the ones used in deciphering wound-induced calcium signaling or ABA or GA signaling.
Author Response
I am grateful for the pointed comments of the reviewer.
Round 2
Reviewer 1 Report
I appreciate the changes made by the author.
Author Response
Thanks for the comments. I have subjected the MS to another round of spellchecking.